# Targeting miRNAs and Other Non-Coding RNAs as a Therapeutic Approach: An Update

**DOI:** 10.3390/ncrna9020027

**Published:** 2023-04-13

**Authors:** Emine Bayraktar, Recep Bayraktar, Hulya Oztatlici, Gabriel Lopez-Berestein, Paola Amero, Cristian Rodriguez-Aguayo

**Affiliations:** 1Department of Gynecologic Oncology, The University of Texas MD Anderson Cancer Center, Houston, TX 77030, USA; 2UTHealth Houston Graduate School of Biomedical Sciences, The University of Texas MD Anderson Cancer Center, Houston, TX 77030, USA; 3Department of Translational Molecular Pathology, The University of Texas MD Anderson Cancer Center, Houston, TX 77030, USA; 4Department of Experimental Therapeutics, The University of Texas MD Anderson Cancer Center, Houston, TX 77030, USA; 5Department of Histology and Embryology, Gaziantep University, Gaziantep 27310, Turkey; 6Center for RNA Interference and Non-Coding RNA, The University of Texas MD Anderson Cancer Center, Houston, TX 77030, USA

**Keywords:** microRNA, miRNA, cancer, drug delivery, resistance, therapeutics

## Abstract

Since the discovery of the first microRNAs (miRNAs, miRs), the understanding of miRNA biology has expanded substantially. miRNAs are involved and described as master regulators of the major hallmarks of cancer, including cell differentiation, proliferation, survival, the cell cycle, invasion, and metastasis. Experimental data indicate that cancer phenotypes can be modified by targeting miRNA expression, and because miRNAs act as tumor suppressors or oncogenes (oncomiRs), they have emerged as attractive tools and, more importantly, as a new class of targets for drug development in cancer therapeutics. With the use of miRNA mimics or molecules targeting miRNAs (i.e., small-molecule inhibitors such as anti-miRS), these therapeutics have shown promise in preclinical settings. Some miRNA-targeted therapeutics have been extended to clinical development, such as the mimic of miRNA-34 for treating cancer. Here, we discuss insights into the role of miRNAs and other non-coding RNAs in tumorigenesis and resistance and summarize some recent successful systemic delivery approaches and recent developments in miRNAs as targets for anticancer drug development. Furthermore, we provide a comprehensive overview of mimics and inhibitors that are in clinical trials and finally a list of clinical trials based on miRNAs.

## 1. Introduction

Non-coding RNAs (ncRNAs) consist of diverse classes of RNA transcripts that are not translated into proteins but are known to regulate the transcription, stability, or translation of protein-coding genes in eukaryotic cells [1,2,3]. ncRNAs can be categorized into housekeeping RNAs and regulatory RNAs. Housekeeping RNAs are transfer RNAs (tRNAs), ribosomal RNAs (rRNAs), small nuclear RNAs, and small nucleolar RNAs [4]. Regulatory RNAs can be classified based on their transcript length and include small RNAs and lncRNAs. Small RNAs, which include piwi-interacting RNAs (piRNAs), small interfering RNAs (siRNAs), small nucleolar RNAs (snoRNAs), and microRNAs (miRNAs), are characterized by their relatively short length, typically less than 200 nucleotides, while transcripts are commonly referred to as lncRNAs (long non-coding RNAs) when they exceed 200 nucleotides in length [5,6].

Among the ncRNA classes, miRNAs and lncRNAs have received the most attention [7,8,9]. miRNAs are small endogenous ncRNA molecules, usually 18–22 nucleotides long, that regulate many developmental and physiological processes, and their role in many health disorders has been strongly supported in the literature over the past two decades [10,11,12]. Several hundred miRNAs have been identified in higher eukaryotes, and it has been demonstrated that they are highly conserved across species and generally function to negatively regulate coding and non-coding gene expression at the post-transcriptional level [10]. In 1993, lin-4 miRNA was the first miRNA discovered. Two research groups led by Ambros and Ruvkun independently reported that the small non-protein-coding RNA lin-4 was found to regulate the expression of the lin-4 gene through its 3′-untranslated region (3′UTR) [11,12,13]. Five years later, Fire and colleagues demonstrated the mechanism of RNA interference (RNAi) in the nematode Caenorhabditis elegans when showing the activity of double-stranded RNAs (dsRNAs) in the activation of the RNAi mechanism and the silencing of messenger RNA (mRNA) sequences [14]. From evidence generated following the discovery of this mechanism and that of let-7, the first miRNA discovered in mammalians, RNAi is now believed to exist in all animals [15,16].

The initial evidence that miRNAs are involved in human cancer was provided by Croce, Calin, and colleagues, who identified a tumor suppressor at chromosome 13q14 [17]. In these studies, they showed that the aforementioned region is frequently deleted in patients with chronic lymphocytic leukemia, and two miRNA genes, miR-15a and miR-16-1, were found to be encoded within this region. These miRNA genes are deleted or transcriptionally downregulated in hematological malignancies, including chronic lymphocytic leukemia [17]. Further studies revealed that both miRNAs function as tumor suppressors that induce apoptosis by repressing an anti-apoptotic protein named B-cell lymphoma 2 (Bcl-2), which is overexpressed in hematological malignancies [18,19]. This fact was reiterated in experiments, where it was shown that the deletion of a tumor suppressor miRNA cluster in mice recapitulated B-cell-malignancy-associated phenotypes presented in humans, which provides convincing evidence for their tumor suppressor functions [20]. The functional validation of miRNAs in vitro and in vivo has allowed a better comprehension of physiological and pathophysiological processes in normal development and human diseases at the molecular level [5,21,22,23,24,25,26,27]. These studies uncovered a previously undescribed mechanism of post-transcriptional regulation that is highly dysregulated in tumoral cells [28]. With the help of advanced high-throughput and Clustered Regularly Interspaced Short Palindromic Repeats (CRISPR) techniques, such as expression microarrays, next-generation sequencing (NGS), and single-cell analysis, it has been demonstrated that miRNAs are dysregulated in numerous diseases, including infectious and cardiovascular diseases and particularly in almost all types of human cancer [5,23,25,29,30,31,32]. The expression signatures of abnormally expressed miRNAs can have diagnostic, prognostic, or theragnostic implications [33]. The genome-wide profiling of the miRNome allowed the highly accurate discrimination of different types of cancer and the tissue of origin of poorly differentiated tumors [23,29,30,34].

Other types of ncRNAs with regulatory functions have been found (Table 1). Compared to miRNAs or other small ncRNAs, which are normally less than 200 ribonucleotides long, lncRNAs are more heterogeneous, ranging from 200 to thousands of ribonucleotides [1]. Nowadays, it is recognized that lncRNAs are more finely regulated and specifically restricted to cells compared to mRNAs [35]. They have frequently and evolutionarily preserved functions, secondary structures, and regions of microhomology, despite minimal overall sequence similarity [36,37,38,39]. Increasing evidence has identified lncRNAs as having a role in transcriptional and translational regulation and human diseases [40], particularly in cancer [41].

In this review, we discuss (1) the opportunities for miRNAs and other non-coding RNAs as targets for anticancer drug development; (2) recent insight into the physiological role of miRNAs and chemotherapy resistance; (3) approaches to overcoming anatomical and physiological barriers to delivery; (4) the advanced design of delivery strategies for miRNAs, particularly successful in vivo strategies or strategies that have introduced novel functionalities to enhance the intracellular trafficking of miRNA mimics or inhibitors in cancer therapy; and (5) emergent insight into the role of lncRNAs and their therapeutic opportunities as targets for anticancer therapy.

## 2. Canonical miRNA Biogenesis

RNAi mechanisms are the genome’s strategy against mobile genetic elements such as transposons and viruses, which generate aberrant RNA or dsRNA when active. The initiation of RNAi processing by miRNAs and the prevention of mRNA-to-protein conversion have been well described [58]. Briefly, the endogenous miRNA pathway involves the step-by-step conversion of long primary miRNA transcripts into short dsRNA duplexes that are between 19 and 21 nucleotides long and have 3′ overhangs. In the initial miRNA biogenesis step, the nuclear protein complex, known as a microprocessor, processes long primary miRNA transcripts, pri-miRNAs, into single hairpins designated precursor miRNAs (pre-miRNAs) [59]. The microprocessor complex is formed by the RNase III enzyme Drosha, the double-stranded RNA (dsRNA)-binding protein (dsRBP) DiGeorge critical region 8 (DGCR8, also known as Pasha), and other less well-recognized auxiliary factors, such as the probable ATP-dependent RNA helicase DDX5, also known as DEAD box protein 5 or RNA helicase p68, and the probable ATP-dependent RNA helicase DDX17 (also known as p72) [60,61,62,63]. The pre-miRNAs are exported out of the nucleus to the cytoplasm, mediated by direct interaction with the protein export receptor exportin 5 (XPO5) [64,65,66]. Located in the cytoplasm, the endoribonuclease RNase III enzyme Dicer then catalyzes the further cleavage of this dsRNA, which is 20–25 nucleotides long, from the stem of the pre-miRNA [67] to produce mature miRNAs, which inhibit the translation or cleavage the target transcript [68,69]. Dicer works together with the dsRBP *trans*-activation-responsive RNA-binding protein (TRBP, also known as TARBP2, RISC Loading Complex RNA Binding Subunit). At the end of the process, in a step named the RNA-induced silencing complex (RISC) loading, the double-stranded miRNA created by Dicer is associated with other members of the Argonaute (AGO) protein family. The complex retains the antisense strand because of its thermodynamic stability and discards the other strand (passenger strand), allowing the antisense strand to become the mature miRNA or guide strand. After this step, the activated RISC engages with mRNA transcripts. The mechanisms of action of miRNAs in gene silencing are summarized in Figure 1. When the degree of base pairing between the antisense guide RNA and sense mRNA is substantial but not complete, which is commonly observed in the interaction between mature miRNAs and their target genes, it can lead to either the repression of protein translation or the destabilization of the mRNA via de-capping or de-adenylation mechanisms. If the base pairing is perfect, the mRNA molecule is cleaved, leading to the degradation of the transcript. The interaction between RNA-binding proteins and miRNA processing intermediates, which form part of a broad layer of post-transcriptional regulation of miRNA biogenesis in many tissues and cancer, was recently described [70]. This evidence reveals that even well-understood processes might be far more convoluted than previously presumed and that additional studies are necessary to decipher miRNA biogenesis [71].

## 3. Dysregulation of Mediators of miRNA Biogenesis

A crucial point in the development of cancer is the status of miRNA expression as well as the machinery of miRNA biosynthesis, and this fact is supported by a growing amount of evidence. Several intermediaries of miRNA biosynthesis, including Drosha, Dicer, TRBP, and XPO5, can exhibit tumor suppressor and oncogenic behavior depending on their status and the tumor type in which they are expressed (Figure 1). For example, low or reduced levels of DICER1 or DROSHA mRNA are associated with poor outcomes in human cancers, including lung, breast, skin, endometrial, and ovarian cancers [72]. In addition, it has been shown that the expression of DICER is decreased in some cancers, probably due to induction by hypoxia as a consequence of the epigenetic mechanism of promoter methylation. For instance, low levels of oxygen influence the function of the oxygen-dependent histone demethylases KDM6A and KDM6B as a consequence of their action on trimethylated histone H3 Lys27 (H3K27me3) located on the DICER1 promoter [73].

In contrast to the tumor types described above, metastatic lesions of prostate cancer exhibit high expression levels of DICER1, while advanced-stage cervical cancers and metastasis-prone esophageal cancers with poor patient survival are characterized by the high expression of DROSHA [72]. The extent to which the altered expression of DICER or DROSHA in these tumors results in a global increase in miRNA expression and how this process contributes to carcinogenesis remain poorly understood. It has been reported that somatic mutations in DROSHA and DICER1 in Wilms tumor patients affect miRNA processing as well as global miRNA expression [74,75]. Potentially, oncogenic transcription factors, such as MYC or the RNA-specific deaminase ADARB1, are involved in the regulation of DROSHA expression and affect primary miRNA (pri-miRNA) processing. In addition, both somatic and germline mutations in DICER1 have been linked to cancer syndromes in humans, including familial pleuropulmonary blastoma, pituitary blastoma, and embryonal rhabdomyosarcoma [76,77,78]. XPO5 is mutated in tumors with microsatellite instability, a gene whose protein product is responsible for transporting pre-miRNAs from the nucleus to the cytoplasm [79]. The loss of function of XPO5 causes the protein to be shortened at its C-terminus, which results in an inability to transport pre-miRNAs from the nucleus to the cytoplasm. In addition, the TARBP2 gene, which encodes TRBP, an essential cofactor of DICER, is subject to inactivating mutations in tumors that display microsatellite instability [80]. The inactivation of TRBP leads to unstable DICER and disrupts miRNA biogenesis. Therefore, the altered expression of miRNAs and proteins that control their processing can have contrasting effects on tumorigenesis, either promoting oncogenesis or functioning as a tumor suppressor. Despite this complex mechanism, their recurrent links to cancer prognosis highlight the critical role of miRNA-mediated gene silencing in tumorigenesis.

## 4. The Role of miRNAs in Cancer Therapy Resistance

Conventional chemotherapy faces many challenges due to the occurrence of severe adverse effects and the development of multidrug resistance (MDR) in cancer treatment; therefore, as drug-resistant cancers become more prevalent, it becomes increasingly important to conduct additional research and develop alternative therapeutic strategies [81,82]. In addition, the mechanism of RNAi, which utilizes miRNAs to induce gene silencing at a post-translational level, has emerged as a novel and promising approach to cancer therapy [14,83]. This statement suggests that miRNA-based therapy can enhance the effectiveness of conventional chemotherapy drugs by increasing the sensitivity of cancer cells to treatment by downregulating efflux transporters, silencing anti-apoptotic genes, reversing the epithelial-to-mesenchymal transition, suppressing angiogenesis, or inducing cell death mechanisms [84,85]. In a recent study, researchers used a combination of Let7b with paclitaxel and showed that the therapy significantly decreased MDR gene expression, resulting in elevated antitumor activity in an MDR model of epithelial ovarian cancer. Additional studies revealed that such combination therapy was reliable when administered repeatedly [86]. Normann et al. found that miR-101-5P sensitized HER2+ breast cancer cells to trastuzumab and lapatinib therapy and caused growth inhibition [87]. Therefore, the combination of chemotherapy and iRNA-based therapy is a potent strategy for synergistic effects in cancer therapy through their different mechanisms against target cancer cells [10,14,82,84,85]. Additionally, miRNAs have also demonstrated potential as promising targets for understanding the mechanisms of chemoresistance in different types of cancer [19,20,21]. The number of known involved genes, including miRNAs, and the association between signaling pathways in drug resistance are being rapidly elucidated [88]. Despite the functional characterization of miRNAs in the development of resistance mechanisms, only a few studies have reported on the complex interactions between miRNAs and resistance mechanisms in cancer. In this section, we discuss and give examples of some miRNAs that are involved in the development of resistance in cancer treatment.

5-Fluorouracil (5-FU), which is an anti-metabolite, has numerous therapeutic benefits for treating different kinds of cancer and can be used with other cytostatic agents with minor side effects. However, its effectiveness is limited in colorectal cancer (CRC), which is often unresponsive to other treatments [89]. The primary treatment response is often diminished by the development of resistance to 5-FU, which is usually given in combination with the modulator folinic acid [89,90]. To address this issue, researchers investigated the role of miR-206 in 5-FU resistance and found that it targets Bcl-2 to mediate chemoresistance, proliferation, and apoptosis in CRC [91]. Additionally, in another study, it has been reported that miR-26b tumor suppressor overexpression in CRC cells is linked to low P-gp (P-glycoprotein) expression and increased sensitivity to 5-FU treatment [92]. These studies provide a novel promising candidate for CRC therapy.

Although microtubule-targeting agents (MTAs) are commonly used to treat non-small-cell lung cancer (NSCLC), drug resistance often limits the response rate to about 25%. Yu and colleagues have demonstrated that miR-195 works synergistically with MTAs to suppress NSCLC cell growth in vitro. Increased miR-195 expression sensitizes NSCLC cells to MTAs, while the repression of miR-195 results in resistance to MTAs. The study also revealed that NSCLC tumors with high levels of miR-195 expression are more responsive to MTA treatment, and inducing miR-195 expression in NSCLC tumors can enhance the antitumor effect of MTAs. These results highlight the crucial role of miR-195 in regulating the NSCLC cell response to MTAs and suggest that miR-195 could serve as a biomarker for the response to MTAs and an adjuvant to MTA therapy [93].

Ovarian cancer is a highly lethal type of gynecological cancer, primarily due to its late-stage diagnosis, rapid progression, high heterogeneity, low prevalence of recurrence, and resistance to current chemotherapy drugs. Many studies have reported a correlation between miRNAs and chemoresistance in ovarian cancer [88,94]. BRCA1/2-mutated ovarian cancer is defective in homologous recombination repair (HRR) of double-strand breaks (DSBs) and thereby sensitive to platinum and PARP inhibitors (PARPis). Multiple PARPi have recently received US Food and Drug Administration (FDA) approval for the treatment of ovarian cancers, and resistance to PARPi is a major clinical problem. Using primary and recurrent BRCA1/2-mutated carcinomas from ovarian cancer patients, patient-derived lines, and an in vivo BRCA2-mutated mouse model, Meghani et al. identified that a microRNA, miR-493-5p, induced platinum/PARPi resistance exclusively in BRCA2-mutated carcinomas, and they concluded that the cumulative impact of miR-493-5p on multiple pathways related to genome stability causes PARPi/platinum resistance in BRCA2-mutated carcinomas [95]. Sun et al. found that after the overexpression of miR-506-3p, the β-catenin level and sensitivity to the PARPi olaparib and cisplatin decreased in ovarian cancer cells, and the results were also supported by rescue experiments, suggesting the promising role of miRNAs in overcoming ovarian cancer treatment resistance [96].

Mutations or changes in the expression levels of apoptosis-related genes or proteins, such as tumor suppressor protein p53 (TP53), can lead to drug resistance in cancer. TP53 is responsible for initiating apoptosis in response to chemotherapy, and mutations in this protein are often associated with drug resistance [81,97]. Van Roosbroeck et al. have demonstrated the role of miR-155 in chemotherapy resistance and explored the potential of anti-miR-155 treatment to sensitize lung cancer to chemotherapy. Their findings revealed that miR-155 promotes resistance to multiple chemotherapeutic agents, such as cisplatin and doxorubicin, in vitro. Moreover, the systemic administration of anti-miR-155 was able to sensitize tumors to chemotherapy in vivo, leading to reduced tumor size and metastasis mass. A negative feedback mechanism was shown between miR-155 and TP53, which contributes to resistance to multiple types of chemotherapeutic agents across various types of tumors [98].

Several apoptotic inhibitor proteins, including myeloid cell leukemia-1 (Mcl-1), Bcl-2, and Bcl-xL, have been shown to alter the response to cisplatin [99]. In some cancers, cisplatin resistance is caused by the overexpression of anti-apoptotic Bcl-2 [99,100]. Furthermore, some ongoing clinical trials in several cancers have been including the combination of cisplatin with small molecules, such as ABT-263, that inhibit BCL-2-like proteins [101,102].

A recent study has demonstrated that miR-503 plays a role in regulating the resistance of non-small-cell lung cancer cells to cisplatin treatment. The overexpression of miR-503 sensitized A549 cells to cisplatin, whereas the inhibition of miR-503 in A549 cells increased resistance to cisplatin by targeting the anti-apoptotic protein Bcl-2, which is upregulated in resistant A549 cells. The ectopic overexpression of miR-503 reduced the Bcl-2 protein level and sensitized A549 cells to cisplatin-induced apoptosis. These results suggest that miR-503 regulates cell apoptosis by targeting Bcl-2 and thus modulates the resistance of non-small-cell lung cancer cells to cisplatin [100].

Hepatocellular carcinoma (HCC) is notoriously resistant to standard therapy due to the development of multidrug resistance, including sorafenib [103,104]. To overcome these resistance problems, future studies are urgently needed. Recently, Awan and colleagues have studied the harmonious effect of miR-17-92 cluster inhibitors/mimics and circular RNAs on sorafenib-resistant HCC cells to explore potential resistance mechanisms and to identify putative targets for sorafenib-resistant HCC cells using a hybrid Petri-net-based computational model that integrates seven miRNAs belonging to the miR-17-92 cluster and crosstalk between EGFR and IL-6 signaling pathways, which are differentially regulated by these miRNAs. They found that the critical targets of miR-17-92 involved in sorafenib resistance have synergistic relationships [104]. Additionally, it has been reported that the ectopic expression of miR-19a-3p, a member of the miR-17-92 cluster, contributes to HCC metastasis and sorafenib chemoresistance by modulating the expression of phosphatase and tensin homolog (PTEN) and PTEN-dependent pathways [105].

MDR and disease relapse remain major obstacles in the treatment of acute myeloid leukemia (AML), which is a highly invasive malignant hematopoietic system disease [106,107]. Recently, Wang and colleagues found that the miR-493-5p-dependent suppression of methyltransferase-like 3 (METTL3) increases the sensitivity of AML cells to the chemotherapeutic drug arabinocytosine [108]. There is evidence that miRNAs also contribute to increased chemoresistance in AML. It has been shown that miR-15a-5p and miR-21-5p induced cytarabine and/or daunorubicin chemoresistance by targeting pro-apoptotic genes in patients with cytogenetically normal AML [109]. These studies have revealed the different roles of miRNAs in AML cell resistance.

Malignant plasma cells infiltrate the bone marrow in individuals with multiple myeloma (MM), which is the second-most prevalent and, currently, an incurable type of hematologic cancer [110]. Although there are currently effective therapeutic regimens available for the treatment of MM, drug resistance is still a major obstacle in the treatment of patients. Therapeutic approaches, such as immunomodulatory drugs (e.g., lenalidomide and pomalidomide), proteasome inhibitors (e.g., bortezomib and carfilzomib), monoclonal antibodies, checkpoint inhibitors, and hematopoietic stem cell transplantation, have contributed to recent advances in the treatment of MM. Unfortunately, plasma cells may develop therapeutic resistance before a patient receives therapy or ingests conventional drugs [111]. Therefore, the development of novel therapeutic interventions is highly required to overcome this resistance problem. Rastgoo et al. demonstrated that EZH2 was abundantly expressed in drug-resistant MM cells as well as associated with a poor prognosis in MM patients [112]. This study has identified the crucial involvement of EZH2 overexpression in drug resistance and tumor progression and proposes targeting the EZH2/miR-138 axis as a promising therapeutic approach for MM [112]. Functional studies indicate that targeting EZH2 through the use of miR-138 or EZH2 inhibitors represents a promising therapeutic approach for addressing drug resistance in MM [112].

Breast cancer is the most common malignancy in women [113]. The majority of breast cancers are estrogen receptor alpha-positive (ER^+^α) at primary diagnosis and depend on estrogen for their growth and progression [114]. Endocrine therapies, including tamoxifen, have been used to target estrogenic stimulation of tumor growth. However, ER^+^ breast cancers tend to develop resistance to both endocrine therapy and chemotherapy, resulting in disease recurrence and leading to tumor progression [115]. One study has demonstrated the association between miRNA expression and tamoxifen resistance in breast cancer [115]. miR-21, miR-146a, miR-148a, miR-34a, and miR-27a are differentially expressed in parental MCF-7 and tamoxifen-resistant MCF-7 cells, suggesting that they are potential therapeutic targets in tamoxifen-resistant breast cancer [115]. Another recent study revealed a novel association between miR-663b and TP73 in tamoxifen-resistant ER+ breast cancer [116]. It was demonstrated that the inhibition of miR-663b decreased cell proliferation and triggered cell apoptosis, resulting in enhanced TAM sensitivity in MCF-7 cells by targeting TP73 [116]. A summary of the miRNAs related to tumor resistance is given in Table 2.

### Approaches to Overcoming Anatomical and Physiological Barriers to Delivery

Over the past 10 years, interest in RNA-based treatments has exploded due to their great selectivity for their target RNA or DNA. The human and viral genomes can be specifically targeted with these drugs to downmodulate gene expression and change mRNA splicing. In addition, targeting ncRNAs by RNA-based treatments may impact the expression of specific genes [117]. Although there are many advantages of RNA-based therapies, there are also challenges because of their natural structures.

**Table 2 ncrna-09-00027-t002:** miRNAs involved in the regulation of cancer therapy resistance.

miRNA	Effect on Resistance ^1^	Drug	Cancer Type	Ref.
Let7b	Decreases MDR gene expressions, increases sensitivity	PTX	EOC	[86]
miR-101-5P	Inhibits growth, increases sensitivity	Trastuzumab and lapatinib	BC	[87]
miR-206	Mediates chemoresistance, proliferation, and apoptosis	5-FU	CRC	[91]
miR-26b	Downregulates P-gp, increases sensitivity	5-FU	CRC	[92]
miR-195	Inhibits growth, increases sensitivity	Microtubule-targeting agents (MTAs)	NSCLC	[93]
miR-493-5p	Increases chemoresistance	Platinum/PARPi	BRCA2-mutated carcinomas	[95,108]
miR-506-3p	Decreases β-catenin, decreases sensitivity	PARPi	OC	[96]
miR-155	Induces chemoresistance	Cisplatin, doxorubicin	LC, CLL, ALL	[98]
miR-503	Downregulates Bcl-2, increases sensitivity	Cisplatin	NSCLC	[100]
miR-17-92 cluster	Increases chemoresistance	Sorafenib	HCC	[104]
miR-19a-3p	Increases metastasis, modulates PTEN expression, increases chemoresistance	Sorafenib	HCC	[105]
miR-493-5p	Suppresses METTL3, increases sensitivity	Arabinocytosine	AML	[108]
miR-15a-5p	Targets pro-apoptotic genes, induces chemoresistance	Cytarabine and/or daunorubicin	AML	[109]
miR-21-5p	Targets pro-apoptotic genes, induces chemoresistance	Cytarabine and/or daunorubicin	AML	[109]
miR-138	Targets EZH2, increases sensitivity	Bortezomib	MM	[112]
miR-21	Induces chemoresistance	Tamoxifen	BC	[115]
miR-146a	Increases sensitivity	Tamoxifen	BC	[115]
miR-148a	Increases sensitivity	Tamoxifen	BC	[115]
miR-34a	Increases sensitivity	Tamoxifen	BC	[115]
miR-27a	Increases sensitivity	Tamoxifen	BC	[115]
miR-663b	Increases cell proliferation, induces chemoresistance	Tamoxifen	BC	[116]

^1^ The table shows the effects in the case of related miRNA upregulation.

Evolutionary barriers have created a variety of challenges in the delivery of RNA therapeutics. Some of the challenges are the oligonucleotide size and charge, RNase susceptibility, the reticuloendothelial system, immunogenicity, and endocytosis. However, delivery through lipid bilayers is still the greatest problem among these obstacles. Small-molecule inhibitors typically have a small enough molecular weight (1 kDa) and sufficient hydrophobicity (logP value) to allow them to easily pass through the lipid bilayer of a cell membrane [118]. All RNA-based treatments, in contrast, are massive macromolecules that are unable to pass through lipid bilayers. Their sizes range from 4 to 10 kDa for single-stranded antisense oligonucleotides (ASOs) to 14 to 200 kDa for CRISPR-Cas9 single-guide RNAs to 700 to 7000 kDa for self-replicating mRNAs [117,119].

RNA molecules cannot easily traverse biological membranes because they are negatively charged and hydrophilic, and biological membranes also include a hydrophobic part. Additionally, the endonucleases and exonucleases found in body fluids may quickly break down RNA. The activation of Toll-like receptors, which have developed to recognize microbial infections by detecting extrinsic nucleic acids, can lead to the initiation of the innate immune response by foreign RNA [120,121]. Even though the immune system’s activation may be advantageous in some circumstances, such as in response to vaccinations or immunotherapies, it may be harmful in other situations. Furthermore, because of their short half-lives due to fast degradation and renal clearance, the unfavorable pharmacokinetic profile of RNA therapies might impair their capacity to reach their desired site of action [122].

RNA-based therapeutics are macromolecules that are taken up by endocytosis but are locked inside endosomes, which are encapsulated by a lipid bilayer and thus are absent from the cytoplasm or nucleus. Getting through the endosomal lipid bilayer presents a challenge for all forms of endocytosis, despite the existence of several types, such as clathrin, caveolae, phagocytosis, macropinocytosis, and others [123]. The main technological challenge for RNA-based treatments to reach their full potential is their transportation from the endosome to the cytoplasm [117].

The other challenge to the delivery of RNAs is the blood–brain barrier. The vascular blood–brain barrier prevents oligonucleotides such as RNAs from passively diffusing into the central nervous system. Oligonucleotides without a delivery agent need to be administered directly into the spinal cord or directly into the brain by Intracerebroventricular injection using an Ommaya reservoir (Figure 2). Intrathecal administration is the most popular method for delivering drugs to the central nervous system in humans. Using this method, ONs are injected into the spinal cord’s subarachnoid space, where they can pass through the pia mater and enter the parenchyma. As a result, the concentration of oligonucleotides in the cerebral spinal fluid rises quickly, allowing for the use of lower drug dosage and a consequent decrease in adverse effects [124]. Thus, RNA therapies using this delivery method have proven to be ineffective and insecure.

## 5. Systemic Delivery and Recent Developments Using miRNAs as Targets for Anticancer Drug Development

The fundamental barriers to miRNA- and gene-based therapies include their limited uptake by tumors and the need to achieve the proper localization of the nucleic acids to the cytoplasm and ensure their efficient delivery to the targeted tissues [125]. The systemic delivery of miRNAs in vivo is the most challenging problem due to several barriers that miRNAs have to overcome before reaching their target mRNAs [126]. Despite these problems, replacement or inhibition-based miRNA gene therapy is a promising approach to targeting multiple pathways simultaneously. miRNA/gene-based therapy can generally be divided into viral and non-viral carriers. The utilization of genetically modified viruses enables highly efficient gene silencing and facilitates the expression of multiple copies of RNAi molecules from a single transcript, which is advantageous for viral delivery systems [127]. However, there are major disadvantages that restrict the widespread use of viral components: adverse immune reactions, viral gene insertion into the genome, and expensive production restrain the widespread use of viral vectors [128]. In contrast, non-viral vectors are commonly used for miRNA- and gene-based delivery. While non-viral vectors exhibit lower efficacy in gene silencing, significant advancements in their development have led to their increasing use in miRNA/gene-based delivery with promising outcomes [125,129]. In recent years, non-viral delivery systems have demonstrated considerable advancements in the efficient delivery of molecules that possess high structural and functional tenability. These systems also have the potential to reduce interactions with non-target cells, improve cell entry and endosomal escape, resist renal clearance, and produce low toxicity and immunogenicity [117,130].

Non-viral delivery systems are typically categorized as polymeric vectors, lipid-based carriers (including positively charged, negatively charged, or neutral formulations), and inorganic materials. The following section focuses specifically on lipid-based drug carrier systems.

### 5.1. Lipid-Based miRNA Carrier Systems

Lipid-based miRNA carriers or particle systems are typically smaller than 100 nm in diameter and deliver miRNA into the cytoplasm of tumor cells with the assistance of cationic lipids [131,132]. In order to be successfully administrated in vivo, miRNA mimics or inhibitors must be encapsulated in lipid-based nanoparticles, which can improve their distribution within tissues and enhance their ability to target tumors compared to miRNAs that are not encapsulated [133]. Due to the fact that lipids and phospholipids are the primary components of the cell membrane, lipid-based miRNA carrier systems have a natural affinity for the cell membrane, which allows them to be easily taken up by cells [133]. Molecules such as DNA, oligonucleotides, and miRNAs that are not encapsulated are not particularly stable and can be quickly broken down by enzymes, trapped within structures such as endosomes inside cells, and eliminated from the bloodstream by the human immune system [134,135]. A novel lipid-based miRNA delivery strategy has emerged, which involves incorporating surface modifications to lipid-based nanoparticles. These modifications improve the stability of the nanoparticles in serum, promote their entry into cells, and facilitate their escape from endosomes while also reducing their toxicity and potential to trigger an immune response [134,135]. Several biocompatible and biodegradable lipids and phospholipids are being developed to create lipid-based miRNA carrier systems. Several classes of lipid-based systems have been developed for miRNA delivery, most notably liposomes, solid lipid nanoparticles, and nanostructured lipid carriers.

The section that follows discusses the present obstacles and techniques involved in delivering lipid-based miRNAs to treat various types of cancer.

#### 5.1.1. Liposomes

Liposomes are synthetic vesicles made of phospholipids and can range in size from 50 nm to ≥1 µm. Liposomes are composed of a lipid bilayer of amphipathic lipid molecules that surround an internal aqueous phase [136]. Due to their structure, liposomes have the ability to encapsulate hydrophilic drugs within their inner aqueous compartment and hydrophobic drugs within their lipid bilayer [136,137]. The pharmaceutical industry has developed numerous liposomal products since 1995, including doxorubicin (Doxil^®^), daunorubicin (Daunoxome^®^), amphotericin B (Ambisome^®^), and cytarabine (Depocyte^®^) [136,137].

Liposomes are important delivery systems for protecting nucleic acids from nuclease degradation and are especially useful for miRNA delivery [138]. Positively charged lipids are predominately formed for miRNA delivery by lipoplexes, which occur when the negatively charged miRNAs bind to positively charged lipids [138]. In this section, we discuss some studies that have used different formulations of liposomes for in vivo miRNA delivery in many types of cancer.

1,2-Dioleoyl-sn-glycero-3-phosphatidylcholine (DOPC) is a neutral lipid that has been shown to be non-toxic in vivo, making it a desirable option for miRNA delivery [139]. Our group has demonstrated that the use of DOPC nanoliposomes for miRNA mimic and inhibitor delivery serves as an effective approach [74,75,98,140]. Several in vivo studies have shown that DOPC nanoliposomes deliver siRNAs or miRNAs into tumor cells 10- to 30-fold more efficiently than unmodified siRNAs or miRNAs or cationic liposomes [76,126]. The administration of miRNA mimics or inhibitors using DOPC nanoliposomes via systemic treatment has demonstrated the ability to successfully target the intended tissue in various xenograft models. This treatment approach has shown potential for both increasing and decreasing intratumoral miRNA expression levels, ultimately leading to the regulation of tumorigenesis [74,75,98,140]. Rupaimoole et al. have shown that miR-630 increases tumor growth and metastasis when delivered via a DOPC miR-630 nanoliposome miRNA delivery platform in an in vivo A2780 ovarian cancer model [140]. Tseng et al. demonstrated that systemic miR-200c DOPC nanoliposomes can effectively reach the target tissue, increase intratumoral miR-200c expression, and decrease uterine carcinosarcoma tumor growth in a murine xenograft uterine carcinosarcoma model [74]. Additionally, we studied DOPC liposomal nanoparticles to transport anti-miR-155 for the treatment of lung cancer and found that anti-miR-155-DOPC significantly inhibits miR-155 expression in xenograft lung cancer tumors and that it re-sensitizes chemoresistant tumors to chemotherapy drugs, including cisplatin [98]. We also showed that the replacement of miR-34a via systemically injected nanodelivery-based gene therapy with 1,2-dimyristoyl-sn-glycero-3-phosphocholine (DMPC) and pegylated distearoyl-phosphatidylethanolamine (DSPE-PEG-2000) is safe and inhibits tumor growth in two different models of triple-negative breast cancer [77].

Furthermore, our group has assessed the therapeutic efficacy of miR-6126 and miR-940 in an intraperitoneal orthotopic HeyA8 model of ovarian cancer, and in vivo systemic administration of miR-6126 and miR-940 DOPC nanoparticles significantly suppresses ovarian tumor growth [78,79].

#### 5.1.2. Solid Lipid Nanoparticles

An alternative method for the in vivo delivery of miRNAs is to incorporate solid lipid nanoparticles (SLNs) to improve intracellular delivery and decrease cytotoxicity. SLNs are a newly developed submicron drug delivery system with a mean diameter from 10 to 1000 nm and represent high adaptability, superior handling properties, low toxicity, and better stability in systemic circulation and long-term storage [80,141,142,143,144]. Furthermore, utilizing SLNs for delivery can facilitate the permeability and retention effect, which enables the particles to accumulate in tumor tissues at significantly higher concentrations than in healthy tissues [80,145]. Growing evidence suggests that SLNs are promising for a drug delivery system that provides the sustained release of miRNAs, siRNAs, or other ncRNAs [129,146].

In a study by Liu et al., cationic SLNs were developed for the delivery of miR-200c to the mammospheres of the MCF7 human breast cancer cell line [147]. They demonstrated that the SLN/miR-200c complex provides protection against ribonuclease degradation and that miR-200c-loaded cationic SLNs exhibited a relatively high cellular uptake efficiency [147].

In another study, paclitaxel (PTX) and miR-34a were co-incorporated into SLNs (miSLNs-34a/PTX) to enhance the efficiency of cancer therapy [148]. This co-incorporation of miR-34a and PTX into SLNs resulted in higher nanoparticle uptake by B16F10-CD44+ cells than either miR-34a or PTX alone, and the co-incorporation of miR-34a and PTX enhanced cell death as compared with their separate delivery by SLNs/PTX and miSLNs-34a [148]. In vivo therapeutic administration of miR-34a and PTX through systemic delivery also led to delayed tumor growth in a B16F10-CD44+ tumor model by inhibiting CD44 and inducing apoptosis in B16F10-CD44+ cells [148]. Furthermore, the study demonstrated that miSLNs-34a/PTX has higher cellular uptake efficiency in lung tissues compared with that in other organs, resulting in an increase in their targeted accumulation in tumor-bearing lungs [148]. Additionally, it has been shown that anti-miR-21 oligonucleotides were incorporated into cationic SLNs (named AMO-CLOSs) for silencing upregulated miR-21. The AMO-CLOSs complex was internalized in A549 human lung adenocarcinoma cells, and anti-miR-21oligonucleotides were released from the nanoparticles to target mature miR-21 and subsequently induced apoptosis in A549 cells. Furthermore, AMO-CLOSs showed the high antisense efficiency of miRNA-21 and subsequently reduced the proliferation, migration, and invasion of A549 cells [142].

#### 5.1.3. Nanostructured Lipid Carriers

Despite their advantages, SLNs do have some limitations, such as poor drug-loading capacity, polymorphic transitions, unpredictable gelation tendency, and drug escape during storage [149]. To overcome these effects, nanostructured lipid nanocarriers (NLCs) have been developed as a second generation. NLCs have an aqueous core encircled by a lipid bilayer and a mixed solid and liquid matrix, which makes a more complex matrix that can host more drug molecules compared to SLNs [150,151]. NLCs, as opposed to emulsions, are better able to immobilize medicines and protect particles from coalescing because of the solid matrix. The benefits of SLNs, such as low toxicity, biodegradation, drug protection, gradual release, and the ability to avoid using organic solvents in their manufacture, are likewise advantages of NLCs [151].

There are now three main categories of NLCs based on variations in their lipid and oil contents: Type I (imperfect type), Type II (multiple types), and Type III (amorphous type) [152]. They can also be divided into three types according to their charge: cationic, neutral, and targeting-modified. These NLCs have a promising future for a variety of therapeutic applications due to their excellent biocompatibility, high biodegradability, and low immunogenicity. They have been used in the delivery of nucleic acids, including specific miRNA molecules for tumor gene therapy [150]. Positively charged cationic NLCs can be used for the delivery of negatively charged structures such as polypeptides, oligonucleotides, RNAs, and DNAs. Additionally, cationic NLCs can be used for miRNA delivery for cancer treatment. 

Chen and colleagues used a cationic NLC to deliver miR-34a to examine the experimental lung metastasis of melanoma cells and detected the significant inhibition of cell migration [152]. Another cationic NLC has been tested for the delivery of miR-107 in head and neck squamous cell carcinoma by Piao et al., who found that after delivery with the cationic NLC, the colony formation, invasion, and migration of tumor cells were inhibited in vitro and in vivo [153].

Neutral NLCs have gained interest recently as a new miRNA carrier. Neutral NLCs are not made up of cationic lipids like cationic NLCs are. As a result, many of the drawbacks associated with charge may be avoided using neutral NLCs. For instance, it is difficult for neutral NLCs to cluster in biofluids and avoid being filtered by the liver, adhering to the endothelium, or being ingested by macrophages. Wang et al. reported that neutral NLCs have been used successfully to deliver miRNAs such as miR-124, miR-34a, miR-495, and let-7 in lung cancer and miR-34a in diffuse large B-cell lymphoma [150]. These findings suggest the potential safe use of neutral NLCs because of their low toxicity and non-immunogenicity.

To enhance the target-specific delivery and stability of miRNAs and reduce macrophage recognition, NLCs can be coated with biocompatible polymers such as PEG or ligands such as transferrin [150]. For instance, Chen et. al. has shown that the treatment of experimental lung metastasis of murine B16F10 melanoma involved miR-34a delivery by NLCs modified with a tumor-targeting single-chain variable fragment (scFv). The miR-34a delivered by scFv-targeted NLCs promoted tumor cell death, prevented cell migration, and downregulated survivin while suppressing the mitogen-activated protein kinase (MAPK) pathway [152]. Similarly, in another study, anti-miR-221 was delivered to the human HCC cell line HepG2 using transferrin-modified NLCs, which demonstrated higher effectiveness in delivering anti-miR-221 to HepG2 cells through the transferrin-mediated endocytosis pathway compared to non-targeted NLCs. These findings demonstrate the potential of using ligand-modified NLCs for the targeted delivery of miRNAs [154].

In sum, NLCs can effectively deliver miRNAs to tumor cells depending on the surface charge or the coating ligands. However, further experiments are needed to improve the clinical application of these delivery systems.

## 6. lncRNAs and Their Opportunities as Targets for Anticancer Therapy

Most lncRNAs are biochemically identical to their mRNA counterparts. They are transcribed by RNA polymerase II, have a 5′-cap added, undergo polyadenylation, and are spliced. Despite these similarities, lncRNAs are mostly located in the nucleus and have high levels of cell-, tissue-, and tumor-specific expression. Some lncRNAs are cleaved at their 3′ ends by RNase P, making them more special compared with mRNAs [155].

lncRNAs are implicated in the pathogenesis of cancer. While the majority of the over 60,000 lncRNAs (more than 70%) found in human tumor tissues and cancer cell lines remain poorly annotated, the functional roles of many lncRNAs have been investigated. It is still unclear whether the dysregulation of lncRNAs is a cause or a result of cancer pathogenesis; however, these ncRNAs have added complexity to the molecular pathways of carcinogenesis. Many well-studied lncRNAs behave similarly to protein-coding oncogenes and tumor suppressors that are involved in tumor recurrence and metastasis, including processes such as proliferation, migration, immortality, angiogenesis, and others. Thus, lncRNAs have great potential as targets for cancer treatment and appear to play significant roles in oncogenesis [156,157,158]. Here, we focus on the potential functions of lncRNAs in the development of anticancer drugs.

## 7. The Role of lncRNAs in Cancer Therapy Resistance

Chemotherapeutic drug resistance causes most patient relapses and poor survival outcomes and remains a major issue in cancer therapy despite advancements in the field [159]. lncRNAs have been recognized as crucial molecules for controlling the growth of tumors and as mediators of many chemoresistance mechanisms, including modifying drug efflux, interfering with DNA damage repair, causing apoptosis, and changing drug targets. Additionally, it has been discovered that most lncRNAs enhance chemoresistance, whereas relatively few lncRNAs have an inhibitory effect, in several malignancies, including HCC, ovarian cancer, gastric cancer, breast cancer, and lung cancer [158]. Further studies that provide evidence of chemoresistance mechanisms involving lncRNAs suggest new opportunities to develop effective treatment options in clinical research.

The drug efflux mechanism, mediated by ATP-binding cassette (ABC) transporters, is a significant cause of multidrug resistance (MDR) in many types of cancer. Hydrophobic small-molecule chemotherapy drugs are ejected from cancer cells by ABC transporters, which make up a superfamily of over 48 ABC transporters identified in human genes. P-glycoprotein (ABCB1), ABC efflux transporter protein MRP1 (ABCC1), and breast cancer resistance protein (BCRP) (ABCG2), among other drug efflux transporters, have been identified as important mediators of MDR in cancer cells [160]. Recent studies have focused on the roles of lncRNAs in MDR in different cancer types. Chang et al. found that the lncRNA Linc00518 served as a competitive endogenous RNA and absorbed miR-199a, which controls the production of an ABC efflux transporter in MCF-7 breast cancer cells. As a result of the sponging of miRNA by lncRNA, MRP1 was overexpressed in MCF-7 cells, resulting in MDR. MDR MCF-7 cells were more sensitive to the cytotoxic medicines doxorubicin, vincristine, and paclitaxel when linc00518 was knocked down using the si-linc00518 gene [161]. Huang and colleagues have shown that the lncRNA EPB41L4A-AS2 was downregulated in docetaxel-resistant MDA-MB-231 and MCF-7 breast cancer cells, and lower EPB41LA4-AS2 expression was associated with the upregulation of ABCB1 and the promotion of docetaxel resistance in breast cancer cells [162]. Certain lncRNAs have been shown to control the expression of ABC transporters by changing the concentrations of transcription factors in the nucleus. For example, microarray screening revealed that the lncRNA FOXC2-AS1 is upregulated in docetaxel-resistant osteosarcoma (OS) cells. Additionally, the transcription factor FOXC2 increased ABCB1 to cause OS cells to become chemoresistant [163]. Many lncRNAs upregulate ABC transporter expression and contain metastasis-associated lung adenocarcinoma transcript 1 (MALAT1), which is upregulated in PTX-resistant non-small-cell lung carcinoma; cancer susceptibility candidate (CASC9), which is upregulated in doxorubicin-resistant MCF-7/ADR; and MRUL, which is upregulated in doxorubicin- and vincristine-resistant gastric cancer [163]. These findings indicate that MDR-related inhibition of lncRNAs can be a strategy to decrease chemoresistance in tumor cells.

lncRNAs also can promote chemoresistance by regulating tumor cells’ apoptosis and cell cycle. Guan et al. have revealed that elevated levels of the lncRNA HOTAIR increase the chemoresistance of MM to dexamethasone by regulating apoptosis and cell viability by downregulating the JAK2/STAT3 signaling pathway in peripheral blood, bone marrow samples, and MM cell lines [164]. Similarly, HCC cells that are resistant to chemotherapy drugs, such as docetaxel, 5-FU, and mitomycin, have higher levels of MALAT1 expression. However, when MALAT1 activity is reduced, drug resistance can be reversed. This reduction in MALAT1 activity has been found to decrease the levels of LC3-II, which is involved in autophagy, and increase apoptosis [158,165]. Colon cancer-associated transcript 2 (CCAT2) enhances proliferation and decreases apoptosis in tamoxifen-resistant cells, and the reduction in CCAT2 provides a new approach for breast cancer patients [166]. The lncRNA nicotinamide nucleotide transhydrogenase-antisense RNA1 (lncRNA NNT-AS1) is remarkably expressed in cisplatin-resistant NSCLC tissues and cells, and NNT-AS1 overexpression can change cell proliferation, the cell cycle, and apoptosis through the MAPK/Slug signaling pathway [167]. These findings suggest that investigating the roles of lncRNAs in drug resistance could lead to new treatment strategies for cancer patients. Many lncRNAs have been found to contribute to different types of drug resistance, most of them contributing to tumor growth and metastasis by promoting downstream pathways, and it is necessary to reveal more lncRNAs that can be potential therapeutic targets for chemoresistance in cancer patients. Further research is needed to fully understand the mechanisms.

Plasmacytoma variant translocation 1 (PVT1) is a lncRNA encoded by the human PVT1 gene. This lncRNA is known to regulate tumorigenesis in gastric cancer. Du and colleagues found that PVT1 can modulate the expression of anti-apoptotic Bcl2 and increase 5-FU resistance in gastric cancer. A Kaplan–Meier analysis exhibited that patients who express high PVT1 levels have poor overall survival on 5-FU-based chemotherapy. Instead, treatment without 5-FU chemotherapy can dramatically increase the first progression survival and overall survival of GC patients with high PVT1 expression. The researchers also showed that, in gastric cancer, PVT1 increases 5-FU resistance by activating Bcl-2, which in turn inhibits apoptosis. It may be possible to use PVT1 as a predictor of resistance to 5-FU therapy [168]. Additionally, Chen et al. found that PVT1 regulates cisplatin resistance via the control of autophagy and apoptosis by regulating miR-216b and Beclin-1 in NSCLC cells [169].

Ferroptosis is a newly discovered form of cell death, and it is induced by iron-dependent lipid peroxidation. In a recent study, researchers found a relationship between the lncRNA LINC00239 and CRC. According to the transcriptomic profiles of lncRNAs in primary CRC tissues, LINC00239 was significantly overexpressed in CRC tissues, and LINC00239 has been suggested as a tumor-promoting factor and ferroptosis suppressor in CRC [170].

Pancreatic cancer is one of the most prevalent causes of cancer mortality worldwide due to the absence of early signs, the likelihood of metastases, and the establishment of chemoresistance. Chemotherapy is the most common type of treatment for pancreatic cancer. The chemotherapy drug gemcitabine is often used in the first-line treatment of pancreatic cancer. However, pancreatic cancer often develops drug resistance, negatively impacting prognosis. Therefore, preventing metastasis, detecting biomarkers, and treating chemoresistance are the best ways to increase a patient’s chance of surviving pancreatic cancer. lncRNAs have critical roles in modifying chemosensitivity in pancreatic cancer, according to mounting data [171]. Recent studies have investigated lncRNAs that can decrease, inhibit, or promote gemcitabine resistance in pancreatic cancer. According to these studies, AB209630, cancer susceptibility candidate 2 (CASC2), growth arrest-specific 5 (GAS5), and maternally expressed gene 3 (MEG3) have been found to function as resistance-inhibiting lncRNAs, whereas the HOTTIP, HOTAIR, and PVT1 lncRNAs have been found to function as resistance-promoting lncRNAs in pancreatic cancer [171,172,173,174,175,176,177,178]. All these data provide proof of the potential roles of lncRNAs in drug development for aggressive pancreatic cancer.

In addition to other tumors, lncRNAs have been used to regulate drug sensitivity in ovarian cancer. Lin et al. showed that the lncRNA ACTA2-AS1 was overexpressed in cisplatin-resistant A2780 and SKOV3 cells. Silencing ACTA2-AS1 in cisplatin-resistant ovarian cancer cells reduced cell proliferation [179]. Furthermore, ACTA2-AS1 can sponge the expression of miR-378a-3p, which increased Wnt5a expression. The inhibition of miR-378a-3p counteracted the effect of ACTA2-AS1 knockdown on cell viability. Wang et al. reported that cisplatin-resistant ovarian cancer cells had high expression levels of lncRNA colon cancer-associated transcript 1 (CCAT1). Silencing CCAT1 triggered cisplatin-mediated apoptosis via modulation of Bax, survivin, and Bcl-1 [180].

These findings suggest that lncRNAs may play important roles in regulating drug sensitivity in ovarian cancer and could be potential therapeutic targets for overcoming drug resistance in this disease (Table 3).

To summarize, lncRNAs play crucial roles in regulating chemoresistance in different types of cancer. Targeting specific lncRNAs could potentially reverse chemoresistance in cancer cells. However, there are several issues that need to be addressed before targeting lncRNAs for clinical applications. For instance, it is crucial to verify whether in vivo studies can replicate in vitro experiments regarding the function of lncRNAs in drug sensitivity. Additionally, studies using biological compounds to target lncRNAs need to be further validated before their clinical application in treating cancer patients.

## 8. Clinical Experience

Although small molecules or inhibitors have several advantages, such as high specificity and potency, most cancers are not responsive to these molecules or develop resistance to them. Therefore, more advanced approaches are required to effectively treat these tumors [82]. Recently, the pharmaceutical industry has shifted its focus to ncRNAs as potential drug targets [126]. One promising approach is the use of miRNA mimics or antagonists as cancer therapeutics. This approach has gained attention because of the essential role miRNAs play in regulating gene expression and their involvement in cancer development and progression [82].

In recent years, some miRNA-based delivery mechanisms have been successfully validated using in vivo mouse models and have enabled translation into clinical studies [129]. To date, there are 412 clinical studies investigating miRNA (ClinicalTrials.gov; access date and time: 20 March 2022and 11.20 pm) in many types of diseases, including cancer. Several therapeutic miRNAs are in clinical trials for Hepatitis C Virus infection (miravirsen and RG101), Alport syndrome (RG-102), post-myocardial infarction (MGN-1374), vascular diseases (MGN-2677), cardiac fibrosis (MGN-4220), abnormal red blood cell production diseases such as polycythemia vera (MGN-4893), cardiometabolic diseases (MGN-5804), peripheral arterial disease (MGN-6114), chronic heart failure (MGN-9103), and a variety of cancers (MRX34, miravirsen, and MRG-106) [181].

In this section, we discuss some therapies currently in development or undergoing miRNA-based clinical studies that focus on the involvement of miRNAs in human cancers.

### 8.1. miRNA Mimics in Clinical Trials

miRNAs are frequently deregulated in cancer, leading to the upregulation of oncogenic miRNAs or the downregulation of tumor suppressor miRNAs [182]. Restoring tumor suppressor miRNAs in cancer cells through miRNA replacement has been proposed as a potential therapeutic strategy. This approach involves introducing synthetic miRNA mimics or miRNA expression vectors to cells to restore miRNA function. Several miRNA replacement therapies are currently being explored in preclinical studies and clinical trials, with different approaches including miRNA mimics, miRNA expression vectors, and small molecules that regulate miRNA expression. Table 4 provides an overview of these approaches for miRNA modulation.

The miR-34 family, and miR-34a in particular, is one of the best-characterized tumor suppressor miRNAs and is mainly characterized by downregulation in multiple malignant tumors [77]. Previous studies have shown that the targeted (and inhibited) genes are implicated in tumorigenesis and cancer progression, including FOXM1, WNT, Notch1, CDK4-6, SIRT1, and Bcl-2 [77], and observed complete tumor inhibition in orthotopic mouse models of liver cancer, resulting in no observed immunostimulatory activity or toxicity to normal tissues [183]. Intravenously injected liposome-based miR-34 (MRX34), developed by Mirna Therapeutics, is the first-in-class miRNA replacement therapy for patients with advanced HCC (on ClinicalTrials.gov, identifier: NCT01829971; Table 4). The phase I trial of MRX34 replacement therapy was initiated in May 2013, and preliminary results promised a tolerable safety profile for patients with advanced HCC or melanoma. Unfortunately, during MRX34 replacement therapy, several patients presented with serious immune-related adverse events, resulting in the death of three patients in the trial [184,185,186]. High doses of the miR-34 mimic, at supraphysiological levels, can generate off-target effects on other genes besides the biological target and cause severe immune responses [184,187].

Despite the failure of MRX34, several potential miRNA mimics are still under study in clinical trials and are promising for cancer treatment [186]. TargomiR or MesomiR-1 is loaded with a miR-16 mimic and targets EGFR, which is associated with unsuppressed tumor growth in preclinical models of malignant pleural mesothelioma and NSCLC using a bacterial-derived transfection system, EDV™ nanocells [188]. In this clinical trial, 5 × 10^9^ TargomiRs per week after dose escalation was well tolerated by patients with refractory malignant pleural mesothelioma, and this clinical trial has recently been completed successfully [188]. Despite some adverse complications, this TargomiR trial is expected to continue to phase II [188].

A comprehensive list of ongoing and terminated miRNA replacement therapy clinical trials and profiling studies can be found in Table 4.

### 8.2. miRNA Inhibitors in Clinical Trials

Oncogenic miRNAs, which are overexpressed in most human cancers analyzed to date, inhibit tumor suppressor genes associated with tumorigenesis and cancer development. As a result, the overexpression of oncogenic miRNAs is an attractive therapeutic target, as the downregulation of an overexpressed miRNA holds more potential than the overexpression of a downregulated miRNA [184].

The first miRNA inhibitor in a clinical trial, miravirsen is a short locked nucleic acid against miR-122 developed by Santaris Pharma and has progressed to phase IIa [181]. miR-122 is conserved and highly expressed in hepatocytes and is required for liver homeostasis [189]. As one of the important tumor suppressor miRNAs, hepatic miR-122 is a prognostic biomarker for patients with HCC [189,190]. It has been demonstrated that miravirsen efficiently inhibited hepatitis C viral loads in patients with HCV [191,192]. Another miR-122 inhibitor in a phase IB study, RG101, has a different chemistry than miravirsen by changing its conjugation to an N-acetylgalactosamine structure, increasing its uptake by hepatocytes [192,193].

miR-155 is one of the most conserved and multifunctional miRNAs and is typically overexpressed in solid tumors as well as hematological malignancies, and the overexpression of miR-155 is associated with the tumor subtype, clinicopathologic markers, and poor survival rates in most cancers [30,194,195]. miR-155 is an interesting target for the treatment of cancer because it is present in most cancers at high levels. Reports from several laboratories showed that the inhibition of miR-155 in vitro and in vivo led to increased apoptosis and reduced proliferation, migration, and/or colony formation in different cancers, including osteosarcoma [196], multiple myeloma [197], glioblastoma multiforme [198], endometrial carcinoma [199], lung cancer, chronic lymphocytic leukemia, and acute lymphoblastic leukemia [98].

A synthetic miR-155 inhibitor (locked nucleic acid, LNA anti-miR), MRG-106 (Viridian (formerly miRagen) Therapeutics, Inc., San Diego, CA, USA), is being studied in a phase I clinical trial in patients with cutaneous T-cell lymphoma of the mycosis fungoides subtype, chronic lymphocytic leukemia, diffuse large B-cell lymphoma, and adult T-cell leukemia/lymphoma (ClinicalTrials.gov Identifier NTC02580552) [184].

## 9. Conclusions and Perspectives

ncRNAs are known to play an important role in the development of chemotherapy resistance and disease relapse in many cancers. They can function as either tumor suppressors or oncogenes in various types of diseases. Despite the progress made in clinical trials with miRNAs, such as the successful development of some miRNAs in cancer patients, there are still challenges to be addressed. One issue is the lack of understanding about how ncRNAs regulate coding or non-coding genes. Another challenge is the toxicity of ncRNAs; for instance, high doses of the miR-34 mimic can cause severe immune responses and off-target effects on other genes. To overcome these challenges, new strategies are needed to minimize the toxicity and off-target effects of ncRNAs in patients. Modifying ncRNAs could be a useful approach for personalized therapy that targets resistance mechanisms in certain cancers. Research on ncRNA profiling and the roles of ncRNAs and their targets has become a promising area for drug development or the sensitization of resistant cancer cells to improve the effectiveness of chemotherapy. ncRNAs have become a focus for the pharmaceutical industry as potential drug targets.

## Figures and Tables

**Figure 1 ncrna-09-00027-f001:**
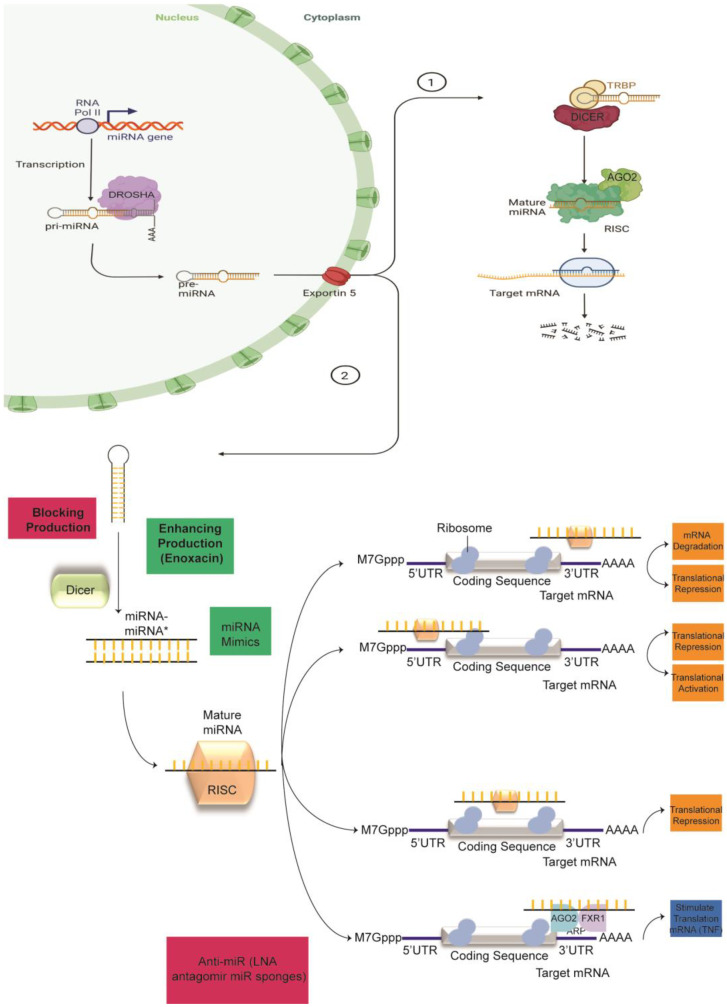
The mechanisms of miRNA biogenesis and overview of points of interference with the endogenous miRNA pathway. miRNA biogenesis is a multistage process that starts with the transcription of a pri-miRNA by RNA polymerase II or III; the pri-miRNA is then processed in the nucleus to a pre-miRNA by the microprocessor complex Drosha and DGCR8. The pre-miRNA is exported by exportin 5 from the nucleus to the cytoplasm. (1) Under normal conditions, pre-miRNA is processed by DICER and TRBP into a mature miRNA duplex. Following the degradation of the passenger strand, the mature miRNA strand is incorporated into the RISC, which modulates gene expression by translational repression or mRNA degradation depending on the level of complementarity with its mRNA target. (2) The inhibition of biogenesis can be carried out in the nuclear or cytoplasmic compartment. Potential points of interference include blocking or enhancing production at the nuclear level, therapeutic miRNA replacement (miRNA mimics), the inhibition (anti-miR) of mature miRNA, or interaction with its target mRNA.

**Figure 2 ncrna-09-00027-f002:**
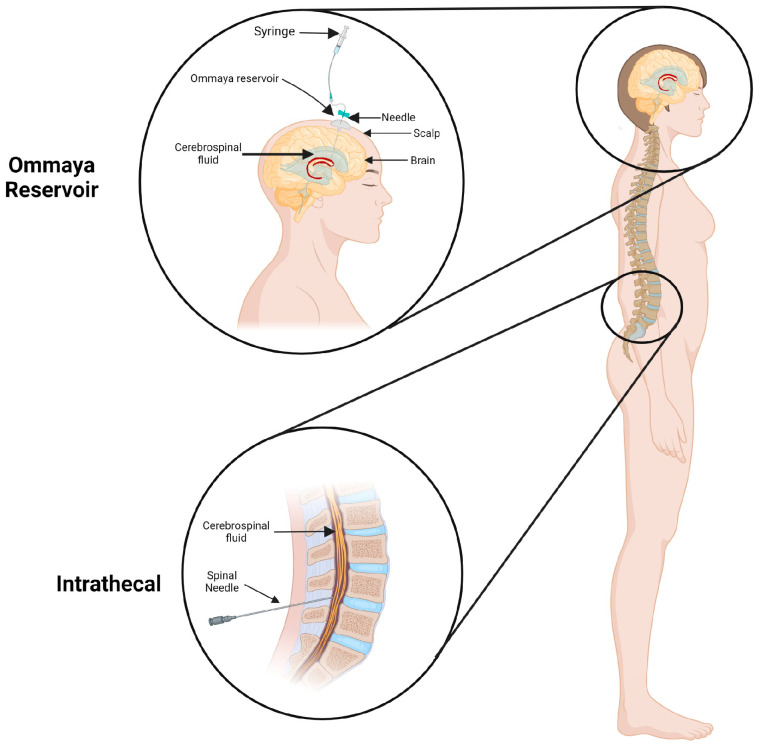
Routes of drug administration into the central nervous system. Intracerebroventricular administration through the Ommaya reservoir into lateral ventricles and intrathecal administration through the spinal column.

**Table 1 ncrna-09-00027-t001:** Cellular non-coding RNAs involved in gene silencing.

Class	Size (Nucleotides)	Functions	Mechanisms	Origin	Model Organisms	Ref.
siRNAs	21–25	Regulate gene expression, provide an antiviral response, and restrict transposons	Degrade RNA, restrict transposon	Intergenic regions, exons, and introns	*Caenorhabditis elegans*, *Drosophila melanogaster*, *Schizosaccharomyces pombe*, *Arabidopsis thaliana*, *Oryza sativa* (rice)	[14]
endo-siRNAs	21–25	Restrict transposons, regulate mRNAs and heterochromatin	Degrade RNA	Transposable elements, pseudogenes	*D. melanogaster*, mammals	[42]
miRNAs	21–25	Regulate gene expression at the post-transcriptional level	Block translation, degrade RNA	Intergenic regions, and introns	*C. elegans*, *D. melanogaster*, *S. pombe*, *A. thaliana*, *O. sativa*, mammals	[11]
piRNAs	24–31	Regulate germline development and integrity and silence selfish DNA	Unknown	Defective transposon sequences and other repeats	*C. elegans*, *D. melanogaster*, *Danio rerio*,	[43]
ra-siRNAs	23–28	Remodel chromatin and silence transcriptional gene	Unknown	Repeated sequence elements (subset of piRNAs)	*C. elegans*, *D. melanogaster*, *S. pombe*, *Trypanosoma brucei*, *D. rerio*, *A. thaliana*	[44]
ta-siRNAs	21–22	Cleave endogenous mRNAs in a *trans*-acting manner	Degrade RNA	Non-coding endogenous transcripts	*D. melanogaster*, *S. pombe*, *A. thaliana*,	[45]
natRNAs	35–150	Regulate gene expression at the post-transcriptional level	Degrade RNA	Opposite strand of the same DNA molecule	All organisms	[46]
scnRNAs	26–30	Regulate chromatin structure	Eliminate DNA	Meiotic micronuclei	*Tetrahymena thermophila* and *Paramecium tetraurelia*	[47]
lncRNAs	200–100	Possibly play roles in epigenetic mechanisms and gene regulation	Degrade RNA and modify epigenetic mechanisms	Intergenic regions, exons, introns	*Drosophila melanogaster,* mammals	[48]
Bacterial riboswitches	Variable	Regulate the gene’s activity in response to the concentration of its target molecule	Block translation	5′-untranslated regions (UTRs) of messenger RNAs	*Bacillus subtilis* and *Neurospora crassa*	[49,50]
snoRNAs	60–300	Guide RNAs in the 2′-O-methylation and pseudouridylation of various classes of RNAs	Involved in nucleolytic processing of ribosomal RNAs to the synthesis of telomeric DNA	U3 species was the first snoRNA described	A broad variety of organisms	[51]
tiRNAs	18	Modulate local epigenetic structure	Not Found	Sequences immediately downstream of the RNA polymerase II transcription start site	Human, chicken, and *Drosophila*	[52]
moRNAs	20	Unknown	Not Found	Located immediately adjacent to miRNAs in the primary miRs	*Ciona intestinalis*	[53]
circRNAs	1500	Regulate gene expression in mammals.	Act as endogenous miRNA sponges and buffer microRNA activity	Originated in different ways (“head-to-tail”, “backsplicing” or reverse splicing)	Human, mouse, and *C. elegans*	[54,55]
lincRNAs	200	Regulate transcription and epigenetic mechanisms	Regulate chromatin topology by both cis and trans mechanisms, the scaffolding of proteins and other RNAs, act as protein and RNA decoys, regulate neighboring genes, and produce micro-peptides	Exonic sequences and promoter regions	*Oryza sativa*	[56]
T-UCRs	200	Unknown	Altered at the transcriptional level in human tumorigenesis; aberrant T-UCR expression profiles may discriminate between different human cancers	Intra- and intergenic regions	Human, mouse, and rat	[37,57]

circRNAs: circular RNAs; siRNAs: small interfering RNAs; endo-siRNAs: endogenous siRNAs; lncRNAs: long non-coding RNAs; lincRNAs: long intergenic non-coding RNAs; miRNAs: microRNAs; moRNAs: miRNA-offset RNAs; natRNAs: natural antisense transcript siRNAs; piRNAs: piwi-interacting RNAs; ra-siRNAs: repeat-associated siRNAs; scnRNA: small scan RNA; snoRNA: small nucleoar RNA; ta-siRNAs: trans-acting siRNAs; tiRNAs: tiny RNAs; T-UCRs: transcribed ultraconserved.

**Table 3 ncrna-09-00027-t003:** lncRNAs involved in the regulation of cancer therapy resistance.

lncRNA	Effect on Resistance ^1^	Drug	Cancer Type	Ref.
Linc00518	Absorbs miR-199a, upregulates MRP1, induces chemoresistance	Adriamycin, vincristine, Paclitaxel	BC	[161]
EPB41L4A-AS2	Downregulates ABCB1, increases chemosensitivity	Docetaxel	BC	[162]
FOXC2-AS1	Upregulates ABCB1, induces chemoresistance	Docetaxel	OS	[163]
MALAT1	Upregulates ABC transporters, increases chemoresistance	PTX, DTX, 5-FU, mitomycin	NSCLC, HCC	[158,163,165]
CASC9	Upregulates ABC transporters, increases chemoresistance	Doxorubicin, gemcitabine	BC, PC	[163,173]
MRUL	Upregulates ABC transporters, increases chemoresistance	Adriamycin, vincristine	GC	[163]
HOTAIR	Downregulates JAK2/STAT3 pathway, increases chemoresistance	Dexamethasone, gemcitabine	MM, PC	[164,177]
CCAT2	Enhances proliferation, increases chemoresistance	Tamoxifen, cisplatin	BC, OC	[166,180]
NNT-AS1	Targets MAPK/Slug pathway, increases chemoresistance	Cisplatin	NSCLC	[167]
PVT1	Modulates Bcl-2 expression, regulates miR-216b and Beclin-1, decreases autophagy and apoptosis, increases chemoresistance	5-FU, cisplatin, gemcitabine	GC, NSCLC, PC	[168,169,178]
AB209630	Targets EphB2 and Nanog, reduces chemoresistance	Gemcitabine	PC	[171,172]
GAS5	Suppresses cell growth and metastasis, increases chemosensitivity	Gemcitabine	PC	[174]
MEG3	Suppresses cell growth and metastasis, increases chemosensitivity	Gemcitabine	PC	[175]
HOTTIP	Downregulates miR-137, increases chemoresistance	Cisplatin	PC	[176]
ACTA2-AS1	Modulates of Bax, survivin, and Bcl-1, increases chemoresistance	Cisplatin	OC	[179]

^1^ The table shows the effects in the case of related lncRNA upregulation.

**Table 4 ncrna-09-00027-t004:** miRNA and terminated replacement therapy and profiling studies in clinical trials.

ClinicalTrials.Gov Identifier	Therapeutic Agent	Target Diseases	Combined Drugs/Therapy	Status
NCT04406831	Serum miRNAs	Pancreatic cancer	-	Recruiting
NCT04305366	miRNA signatures	Squamous cell carcinoma of head and neck	-	Recruiting
NCT04435756	miRNA 371	Germ cell tumors	-	Recruiting
NCT03738319	miRNA signatures	High-grade serous ovarian cancer	-	Unknown
NCT03779022	miRNA biomarkers	Breast cancer	-	Unknown
NCT04972201	Liquid biopsy miRNA	Multi-cancer	-	Recruiting
NCT05495685	Blood miRNAs	Pancreatic cancer	-	Recruiting
NCT04427475	Plasma miRNAs	Advanced non-small-cell lung cancer	Pabolizumab, nafulizumab	Unknown
NCT05556603	Blood miRNAs	Pancreatic cancer	-	Active, not recruiting
NCT04903665	Blood miRNAs	Gynecologic cancer	-	Active, not recruiting
NCT05224596	Blood miRNAs	Gastric cancer	-	Recruiting
NCT03742856	miRNA signatures	Epithelial ovarian cancer	-	Unknown
NCT05431621	miRNA7	Esophageal cancer, gastric cancer, colorectal cancer, hepatocellular carcinoma	-	Recruiting
NCT03742869	miRNA signatures	Uterine cervical adenocarcinoma	-	Unknown
NCT04010487	miRNA expression profiles	Endometrial carcinoma	-	Unknown
NCT04792437	miRNA expression profiles	Glioma	-	Recruiting
NCT05417048	Exosomal miRNA profiling	Breast cancer	-	Not yet recruiting
NCT03397355	miRNA expression profiles	Lung cancer	-	Unknown
NCT01210495	Blood miRNAs	Advanced hepatocellular carcinoma	Axitinib	Completed
NCT03236649	miRNA biomarkers	Advanced hepatocellular carcinoma	Icaritin, sorafenib tosylate	Unknown
NCT03236636	miRNA biomarkers	Advanced hepatocellular carcinoma	Icaritin, HUACHANSU PIAN	Unknown
NCT03108677	Exosomal miRNA profiling	Primary high-grade osteosarcoma	-	Active, not recruiting
NCT03741829	miRNA expression profiles	Small-cell lung cancer	-	Completed
NCT03694483	Blood miRNAs	Prostate cancer	-	Suspended
NCT03526835	miRNA biomarkers	Metastatic colorectal cancer	MCLA-158	Unknown
NCT02509052	miRNA biomarkers	Recurrent plasma cell myeloma, refractory plasma cell myeloma	Leflunomide	Active, not recruiting
NCT04720430	miRNA signatures	Hepatocellular carcinoma	-	Recruiting
NCT03886571	Exosomal miRNA profiling	Pancreatic cancer	-	Recruiting
NCT00926640	miRNA profiling	Small-cell lung carcinoma, malignant epithelial neoplasms	Belinostat, cisplatin, etoposide	Completed
NCT04515004	miR-19a	Early-stage lung cancer	Leucoselect phytosome	Not yet recruiting
NCT03451383	Blood or saliva miRNAs	Breast cancer	-	Recruiting
NCT02724202	miRNA profiling	Colon cancer	Curcumin, 5-FU	Unknown
NCT03770468	miRNA profiling	Glioblastoma	-	Active, not recruiting
NCT04453046	miRNA profiling	Squamous cell carcinoma of the head and neck	Pembrolizumab	Terminated
NCT04158635	miRNA profiling	Pancreatic cancer	Bosentan, gemcitabine, Nab-paclitaxel	Recruiting
NCT02657005	miRNA profiling	Ewing sarcoma	TK216	Terminated
NCT01132586	miR-181	Acute myeloid leukemia	Lenalidomide, cytarabine, idarubicin	Completed
NCT03824327	Hypoxia-inducible miRNAs	Non-small-cell lung carcinoma	Papaverine hydrochloride	Recruiting
NCT01999972	Circulating miRNAs	Advanced solid tumors	Axitinib, crizotinib	Completed
NCT05275075	miRNA expression	Resectable pancreatic adenocarcinoma	-	Recruiting
NCT02507765	miRNA expression	Liver cancer	-	Completed
NCT03824145	miRNA signatures	Breast cancer	-	Recruiting
NCT02642965	miR-29b, miR-499	Acute myeloid leukemia	Cytarabine, fludarabine phosphate	Active, not recruiting
NCT03953443	miRNA expression	Head and neck squamous cell carcinoma	-	Active, not recruiting
NCT03443908	miRNA expression	Lung cancer	-	Withdrawn
NCT01780662	miRNA expression	Hodgkin lymphoma	Brentuximab vedotin, gemcitabine hydrochloride	Completed
NCT03233724	miRNA expression profiles	Non-small-cell lung cancers, Esophageal carcinomas	Decitabine, tetrahydrouridine, pembrolizumab	Recruiting
NCT01050296	miRNA expression	Pediatric solid tumors	-	Recruiting
NCT00681512	Serum miRNA profiles	Non-small-cell lung cancer	Berry Powder	Terminated
NCT02594202	miRNA sequencing	Prostate cancer	-	Recruiting
NCT01676805	miRNA sequencing	Lymph cancer	-	Recruiting
NCT02983279	miR-21	Breast carcinoma, endometrial carcinoma, prostate carcinosarcoma	-	Completed
NCT04100811	miRNA profiling	Prostate cancer	-	Recruiting
NCT04697576	Circulating miRNAs	Stage I, II, and IV melanoma	Ipilimumab, nivolumab, pembrolizumab	Recruiting
NCT01555268	miRNA expression	Acute myeloid leukemia	Trebananib, cytarabine	Completed
NCT05136846	miRNA biomarkers	Stage II-III non-small-cell lung cancer	Carboplatin, durvalumab, paclitaxel	Recruiting
NCT00898092	miRNA expression	Acute myeloid leukemia	-	Active, not recruiting
NCT01629498	Blood miRNAs	Stage II-IIIB non-small-cell lung cancer	-	Recruiting
NCT01446809	Plasma miRNAs	Soft tissue sarcoma	Doxorubicin hydrochloride, ifosfamide	Completed
NCT01583283	microRNA expression profiles	Multiple myeloma	ACY-1215, lenalidomide, dexamethasone	Completed
NCT02323607	miRNA expression	Acute myeloid leukemia	Pacritinib, cytarabine, daunorubicin hydrochloride	Completed
NCT03537599	Exosomal miRNAs	Acute myeloid leukemia	Daratumumab	Completed
NCT01955499	Serum miRNAs	B-cell non-Hodgkin lymphoma	Ibrutinib, lenalidomide	Active, not recruiting
NCT01598285	miRNA signatures	Breast cancer	Bevacizumab	Terminated
NCT01612871	miRNA signatures	Breast cancer	Tamoxifen or anti-aromatase	Completed
NCT02635087	miR-21, miR-20a-5p, miR-103a-3p, miR-106b-5p, miR-143-5p and miR-215	Colon cancer	-	Recruiting
NCT02247453	24 previously identified miRNAs	Lung cancer	-	Active, not recruiting
NCT02812680	Circulating microRNAs	Esophageal adenocarcinoma	Multiregimen chemotherapy	Active, not recruiting
NCT02466113	miR-21, miR-20a-5p, miR-103a-3p, miR-106b-5p, miR-143-5p and miR-215	Colon cancer	Adjuvant chemotherapy	Not yet recruiting
NCT01722851	Circulating microRNAs	Breast cancer	Adjuvant chemotherapy	Completed
NCT02656589	Plasma miRNAs	Breast cancer	Capecitabine and trastuzumab	Unknown
NCT01231386	miRNA signatures	Breast cancer	Neoadjuvant or adjuvant treatment	Completed
NCT03452514	miRNA signatures	Lung cancer	-	Completed
NCT02253251	KRAS-variant and microRNA binding site mutation testing	Breast cancer	-	Recruiting
NCT03293433	miRNA signatures	Pulmonary cancer	-	Completed
NCT01964508	miRNA signatures	Thyroid cancer	-	Completed
NCT02009852	miR-29b	Oral squamous cell carcinoma	-	Unknown
NCT00806650	Serum miRNA signatures	Kidney cancer	-	Completed
NCT01220427	miRNA expression profiles	Prostate cancer	-	Terminated
NCT02366494	Exosomal miRNA profiling	Prostate cancer	Bicalutamide, leuprolide, goserelin, triptorelin, docetaxel	Active, not recruiting
NCT00849914	miRNA signatures	Epithelial skin cancer	-	Completed
NCT03432624	MicroRNA-25	Pancreatic cancer	-	Unknown
NCT02445924	miRNA signatures	Non-small-cell lung cancer	-	Completed
NCT02065908	Circulating microRNAs	Breast cancer	-	Completed
NCT02950207	miRNA-100	Breast cancer	-	Unknown
NCT02127073	miRNA signatures	Breast cancer	Intranasal oxytocin	Suspended
NCT03051191	Expression pattern of miRNA in blood	Cancer and cardiovascular diseases	-	Completed
NCT01143311	miRNA expression profiles	Cutaneous squamous cell carcinoma	-	Terminated
NCT03309722	miRNA expression profiles	Colorectal cancer	-	Recruiting
NCT01712958	miRNA expression profiles	Colorectal carcinoma	-	Unknown
NCT02471469	miRNA expression profiles	Metastatic castration-resistant prostate cancer	Enzalutamide	Completed
NCT01829971	MRX34	Primary liver cancer, Sclc, lymphoma, melanoma multiple myeloma, renal cell carcinoma, Nsclc	-	Terminated
NCT02758652	miRNA expression profiles	Ovarian cancer	-	Active, not recruiting
NCT01541800	Circulating microRNAs	Pediatric cancers: leukemia, lymphoma, central nervous system	-	Unknown
NCT02464930	miR-192-5p, miR-215-5p miR-194-5p	Barrett’s esophagus, gastroesophageal reflux, esophageal adenocarcinoma	-	Unknown
NCT02964351	Circulating microRNAs	Prostate carcinosarcoma	-	Unknown
NCT03338712	miRNA expression profiles	Prostate cancer, radical prostatectomy	-	Withdrawn
NCT02791217	miRNA expression profiles	Lymphoma, B-cell, follicular lymphoma, Hodgkin lymphoma, multiple myeloma	-	Unknown
NCT03074175	Plasma miRNA profiles	Advanced non-small-cell lung cancer	Radiotherapy	Unknown
NCT02531607	Blood miRNAs	Pancreatic neoplasms	-	Active, not recruiting
NCT01240369	miR-326	Non-small-cell lung cancer, esophagus squamous cell carcinoma	-	Unknown
NCT01119573	miRNA profiling in tissues	Endometrial cancer	-	Unknown
NCT02634502	miRNA expression profiles in serum	Pancreatic cancer with liver metastasis	Drug: S-1	Unknown
NCT02268734	Circulating miRNAs	Metastatic sporadic medullary thyroid cancer	Vandetanib	Completed
NCT01595139	miRNA expression profiles	Glioma, neurofibromatosis Type 1	-	Completed
NCT01572467	miRNA expression profiles	Ovarian or testicular sex cord stromal tumors	-	Completed
NCT01391351	miRNA expression profiles	Ovarian carcinoma, fallopian tube cancer, peritoneal, serous-type advanced stage	Taxol and carboplatin therapy and Taxol and carboplatin chemotherapy with avastin	Completed
NCT02364154	Circulating miRNAs	Colorectal cancer	-	Completed
NCT01849952	MicroRNA-10b	Gliomas	-	Recruiting
NCT01927354	MicroRNA-29 family	Head and neck squamous cell carcinoma	-	Unknown
NCT02402036	Serum miRNA profiles	Colorectal cancer	Regorafenib	Terminated
NCT03048266	Serum miRNA profiles	Multiple endocrine neoplasia Type 1	-	Recruiting
NCT00909350	miRNA expression profiles	Barrett’s esophagus, esophageal adenocarcinoma	-	Completed
NCT01595126	Saliva and serum miRNA profiles	Head and neck cancer	Dietary Supplement	Unknown
NCT01828918	miRNA expression profiles	Colorectal cancer	-	Unknown
NCT01965522	Serum miRNA profiles	Breast cancer	Vitamin D and melatonin	Completed
NCT01498250	miRNA expression profiles	Basal cell carcinoma	-	Completed
NCT01500954	miRNA expression profiles	Cutaneous squamous cell carcinoma	-	Completed
NCT01453465	miRNA expression profiles	Brain and central nervous system tumors, kidney cancer	-	Withdrawn
NCT03253107	miRNA expression profiles	Gastric cancer	XP (xeloda + cisplatin) or Xelox (xeloda + oxaliplatin) +/− Herceptin	Recruiting
NCT03202810	Serum miRNA profiles	Oral cancer	-	Unknown
NCT03081988	Circulating miRNAs	Esophageal cancer	Concomitant chemoradiotherapy	Recruiting
NCT01556178	miRNA expression profiles	Hydrocephalus	-	Completed
NCT02807896	miRNA expression profiles	Pancreatic cancer, bile duct cancer, stomach cancer, colon cancer	-	Completed
NCT03429530	Circulating miRNAs	Hepatocellular carcinoma	-	Completed
NCT00581750	miRNA expression profiles	Breast cancer, lobular carcinoma, invasive breast cancer	-	Completed
NCT01970696	miRNA expression profiles	Ovarian stromal tumor, testicular stromal tumors, ovarian small-cell carcinoma, Dicer1 Syndrome	-	Recruiting
NCT03227510	Circulating miRNAs	Hepatocellular carcinoma	-	Unknown
NCT02928627	Hepatic and circulating miR-221 and miR-222	Hepatocellular carcinoma	-	Unknown
NCT03255486	miRNA expression profiles	Advanced breast cancer	-	Completed
NCT01229124	miRNA expression profiles	Acute myeloid leukemia	-	Completed
NCT03362684	miR-31-5p and miR-31-3p	Colorectal cancer	Cetuximab and FOLFOX	Completed
NCT03167476	miRNA expression profiles	Lymphoma, reactive hyperplasia lymphoid	-	Unknown
NCT02507882	miRNA expression profiles	Hepatocellular carcinoma	-	Unknown
NCT01957332	miRNA expression profiles	Metastatic breast cancer	-	Active, not recruiting
NCT00897234	miRNA expression profiles	Lung cancer	-	Completed
NCT01528956	miRNA expression profiles	Pediatric adrenocortical tumors	-	Completed
NCT02862145	MRX34	Melanoma	Dexamethasone	Withdrawn
NCT01345760	miRNA expression profiles	Basal cell carcinoma, squamous cell carcinoma	-	Completed
NCT00743054	miRNA expression profiles	Renal cell carcinoma	-	Completed
NCT01444560	miRNA expression profiles	Cutaneous melanoma	-	Completed
NCT01298414	AML-specific miRs (miR-34a, miR-538e, miR-193e, and miR-198)	Acute myeloid leukemia	-	Completed
NCT01057199	microRNA-34a and microRNA-194	Acute myeloid leukemia	-	Completed
NCT02392377	miRNA expression profiles	Esophageal adenocarcinoma	Paclitaxel, carboplatin, oxaliplatin, leucovorin, calcium, fluorouracil	Terminated
NCT02412579	miRNA expression profiles	Hepatocellular carcinoma	-	Completed
NCT02369198	miR-16 family	Malignant pleural mesothelioma, non-small-cell lung cancer	TargomiR	Completed
NCT01433809	miRNA expression profiles	Thyroid cancer	-	Completed
NCT02448056	miRNA expression profiles	Hepatocellular carcinoma	Sorafenib	Not yet recruiting
NCT03142633	miRNA expression profiles	Polycystic ovary syndrome	-	Completed
NCT01482260	miRNA expression profiles	Cutaneous malignant melanoma	-	Completed
NCT03000335	miR-451, miR-151-5p and miR-1290	Acute lymphoblastic leukemia	-	Unknown
NCT00536029	miRNA expression profiles	Melanoma	-	Completed
NCT00862914	miRNA expression profiles	Malignant melanoma	-	Completed
NCT03474614	miRNA expression profiles	Cerebral cavernous malformations	Propranolol	Unknown
NCT01247506	miRNA expression profiles	Hepatocellular carcinoma	-	Unknown
NCT01505699	miRNA expression profiles	B-cell acute lymphoblastic leukemia	-	Completed
NCT01511575	miRNA expression profiles	Down syndrome acute myeloid leukemia	-	Completed
NCT01606605	miRNA expression profiles	Diffuse large B-cell lymphoma		Completed
NCT00639054	miRNA expression profiles	Multiple myeloma	-	Completed
NCT03416803	miRNA expression profiles	Hepatocellular carcinoma	Radiotherapy	Unknown
NCT01560195	miRNA expression profiles	Advanced non-small-cell lung cancer	Pegylated rhG-CSF	Unknown
NCT02580552	miR-155	Cutaneous T-cell lymphoma, mycosis fungoides, chronic lymphocytic leukemia, diffuse large B-cell lymphoma, ABC subtype adult T-cell leukemia/lymphoma	Cobomarsen (MRG-106)	Completed

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
