# Peer review of "Targeting miRNAs and Other Non-Coding RNAs as a Therapeutic Approach: An Update"

_ncrna, 2023, doi:10.3390/ncrna9020027_

Round 1
Reviewer 1 Report
The manuscript "MicroRNAs and other non-coding RNA as Targets for Anticancer Drug Development" is focused on the physiological role of miRNAs in tumorigenesis and resistance, summarized the approaches to overcoming anatomic and physiologic barriers to delivery and recently advanced design of delivery strategies of miRNAs, provided a comprehensive list of ongoing miRNAs in clinical trials. The authors also discussed the role of lncRNA and their therapeutic opportunities as targets for anticancer therapy. Overall, this review covers broad contents of noncoding RNAs in the cancer therapy with focusing on miRNAs and lncRNAs. The manuscript could be further improved by addressing some minor issues:
1. Figure 1 and Figure 2 could be combined as they show the same pathway of miRNA biogenesis.
2. Figure 2 needs to be illustrated more clearly by either highlighting the points in the figure, such as 1, 2, 3, etc, or explain more in the figure legend, such as pointing out the colors showing in the figure.
3. In section "1.3. The role of miRNAs in cancer therapy resistance", it will be very helpful if the authors could provide a summary table of the miRNAs mentioned in this section, highlighting their functions, advantages, and disadvantages.
4. In section "1.4 Approaches to overcoming anatomic and physiologic barrier to delivery", the authors may need to adjust the content a little bit by putting more words on approaches rather than on the challenges. The current version is more focused on the challenges, not approaches.
5. The authors may need to organize the manuscript in a more logical way by numbering each section carefully. For example, based on the content, section 1.6 is under section 1.5, thus it is more appropriate to number 1.6 as 1.5.1. This is true for other sections, please check the logic between sections and number them appropriately.
6. In section "1.8. The role of lncRNAs in cancer therapy resistance", it will be helpful if the authors could provide a summary figure or table.
7. Line 107, "focusing in successful strategies ...", should be "focusing on successful ...".
8. Line 421, "Recently, non-viral delivery systems ...", please provide references for this.
9. Title "non-coding RNA" should be "non-coding RNAs". Please go through the whole manuscript carefully, to check the word "RNA" whether it should be in singular or plural.
10. Please set italic for "in vivo".
Reviewer 2 Report
This is a timely and very comprehensive review on the therapeutic miRNAs and lncRNAs. It incorporates a number of topics that are not typically covered in noncoding RNA review articles such as drug delivery, effects on the therapeuric efficacy of small moelces. The review does a great job in updating the clinical trials of antmiRs and miR mimetic. Only 2 minor comments for improvement
The print of Fig 1 should be increased for visability
Therefore, the combination of chemotherapy and miRNA-based therapy is a potent strategy line 218
Reviewer 3 Report
In this manuscript, Bayraktar et al. have explored the role of non-coding RNAs as a therapeutic option for cancer. This is an interesting manuscript with several important features related to non-coding RNAs, their role in pre-clinical research, and possible applications in clinical settings. Most importantly, the clinical trial data presented in Table 2 is of great interest to numerous scholars and students. The authors deserve credit for their extensive efforts!
Here are my observations:
- The authors should carefully review the manuscript for English, structure, and paragraph structure. For example, some paragraphs are overly long and should be broken up; sentences should be rearranged to improve the overall clarity of the article. This would significantly enhance the flow of the review.
- In Table 1, the authors can add model organisms instead of “Organisms found in”.
- The authors should refine the introductory paragraph; the introduction of non-coding RNA was abrupt.
- The authors should end the discussion and conclusion section with the main point of the article. The current concluding sentences are too abrupt. “The pharmaceutical industry has been focusing on non-coding RNAs as drug targets.”
- Some of the sentences that need improvement are below. Note that this is not an exhaustive list; authors should look for other sentences that require improvement on their own.
- “In 1993, for the first time, a miRNA was described: lin-4 miRNA was discovered.”
- “Reported in the same issue of Cell (53).”
Round 2
Reviewer 3 Report
The authors have made significant improvements to the manuscript.